# Attitudes and perceptions of undergraduate nursing students towards the nursing profession

**Irene Mildred Neumbe**[1], **Lydia Ssenyonga**[1], **David Jonah Soita**[2], **Jacob Stanley Iramiot** [3]*, **Rebecca Nekaka**[2]

1 Department of Nursing, Busitema University Faculty of Health Sciences, Mbale, Uganda, 2 Department of Community and Public Health, Busitema University Faculty of Health Sciences, Mbale, Uganda, 3 Department of Microbiology and Immunology, Busitema University Faculty of Health Sciences, Mbale, Uganda

* jiramiot@gmail.com

## Abstract

### Background

Nursing students either possess positive or negative attitudes and perceptions about the nursing profession. Their attitudes towards the profession depict the quality of care they will offer to patients upon qualification. This study aimed to determine the attitudes and perceptions of undergraduate nursing students towards their profession.

### Methods

This equal-status mixed methods study involved a census sample of 165 nursing students from year one to year four in two public universities in Uganda. Attitude Scale for Nursing Profession was used to collect quantitative data in the period between 20/11/2021 and 22/01/2022. Focus group discussions were held to collect qualitative data about the perceptions.

### Results

Majority of the students (81.8%) had positive attitudes towards the nursing profession. There was a significant difference in attitudes based on year of study and entry scheme ($R^2$ = 0.12, F = 2.21, p = 0.01). Nursing was perceived as a poorly remunerated, unpopular profession with bias towards recruitment of lower cadres.

### Conclusion

The results of this study showed that the attitudes of undergraduate nursing students towards the nursing profession were positive but their perceptions about the nursing profession were negative. An interventional study is recommended to facilitate a more positive change among nursing students.

**Data Availability Statement:** The datasets generated and/or analysed during the current study are available in the OSF repository, https://doi.org/10.17605/OSF.IO/TXDC6.

**Funding:** Research reported in this publication was supported by the Fogarty International Center of the National Institutes of Health, U.S. Department of State's Office of the U.S. Global AIDS Coordinator and Health Diplomacy (S/GAC), and President's Emergency Plan for AIDS Relief (PEPFAR) under Award Number 1R25TW011213. The content is solely the responsibility of the authors and does not necessarily represent the official views of the National Institutes of Health. The funders had no role in study design, data collection and analysis, decision to publish, or preparation of the manuscript.

**Competing interests:** The authors declare no competing interests.

**Abbreviations:** ASNP, Attitude Scale for Nursing Profession; IRB, Institutional Review Board; MRRH, Mbale Regional Referral Hospital; NCHE, National Council for Higher Education; NGO, Non-Governmental Organization; UNMC, Uganda Nurses and Midwives Council; WHO, World Health Organization.

## Background

Although nurses constitute the largest group of healthcare providers, there is a worldwide suffering from the shortage of nurses to meet the demand of healthcare settings [1].

The World Health Organization International Council of Nurses discussion paper in 2020 estimated the global shortfall of nurses to have been 5.9 million in 2020 and projected it would rise to 10.6 million by 2030 [2]. The shortfall is most severe in Africa, South East Asia, and the Eastern Mediterranean [3]. Uganda is challenged by nursing shortage, especially at graduate level. Certificate and diploma nurses form a large proportion of the nursing workforce in Ugandan healthcare facilities, with only 3.24% of the total nursing workforce comprising graduate nurses [4].

The Nursing profession provides holistic care to the entire population through advocacy, disease prevention, health promotion, health education, counseling, collaboration, research and administration [5]. Despite their enormous contribution, nurses are not adequately recognized. They are exposed to difficult working conditions resulting in stress and fatigue [6]. It is estimated that 30% of the nurses do not like the nursing profession because of low pay, bad administrative system, and work overload [7].

Resolving nurses' attitudes towards the profession is an essential aspect of understanding the issue of nursing shortage since nursing education is negatively affected by negative attitudes towards the profession [7, 8]. Nursing students are the upcoming contributors to the nursing workforce [9] and their attitudes towards the profession depict the quality of care they will offer to patients upon qualification [10].

Nursing students either possess positive or negative attitudes [6], which change over time during their study. At Hue University in Vietnam, over half (62.07%) of the students had a favorable attitude towards the nursing profession [11]. However, in South Valley University Egypt, over half (57.9%) of the students had negative attitudes towards nursing [12]. It has been demonstrated that attitudes of nursing students towards the nursing profession can be affected by gender [12, 13], year of study [14], family support [15], observation of senior nurses [16] and personal experiences with nurses [17].

Nursing students' perceptions about the nursing profession are based on personal experiences and observation of nurses' behavior, however superficial. The profession is poorly perceived yet the job is very hard [18]. Outgoing nursing students in Punjab, India had divergent perceptions of the profession. The majority (81.9%) perceived that there is opportunity for personal growth in the profession, while 23.9% disagreed. Nearly half (51.5%) believed that the profession does not provide any opportunity to get a better marriage partner but provides better opportunities abroad (86.8%) [19]. At the University of Jos, Nigeria, the nursing profession was perceived as stressful but offers good pay, good job security and self-actualization [20].

Nurses are viewed as subordinates of the physician, an inferior profession whose destiny is to obey the doctor's orders [21, 22]. In ancient societies, nursing was a female role that required no formal education. Organized groups of nurses came into existence in the early Christian era [23]. Reforms in nursing were introduced by Florence Nightingale in the nineteenth century. In Uganda, nursing training was started by Katherine Cook in 1931 at enrolled level and the first class qualified in 1933 [5].

The Uganda Nurses and Midwives Council (UNMC) is responsible for regulating nursing and midwifery standards as governed by the Nurses and Midwives Act, 1996. An individual holding a qualification recognized by UNMC applies for registration, upon which they are registered and offered a practicing license. Such a person can then be referred to as a Registered Nurse [24].

The Bachelor of Science in Nursing, introduced in 1993, is a 4-year program in nursing schools affiliated with universities in Uganda. At Busitema University and Makerere

University, nursing students take the same basic courses as medical students during the first two years [25], mainly taught by medical doctors rather than nurse educators. During the final two years of full-time clinical placement, nursing staff in the teaching hospitals, most of whom are diploma-level nurses, serve as clinical instructors for the students [26].

Ugandan universities enroll students according to the students' applications and its enrolment plan. Students with high scores in the Uganda Advanced Certificate of Education examination have more opportunities of enrolment. Those who do not meet admission requirements because of low scores end up applying to other universities or unpopular courses [27]. Therefore, most of the students who study nursing do so after failing to make it to the popular medicine and pharmacy courses.

Nursing training institutions experience pressure from healthcare organizations and communities to increase student enrollment and graduate more nurses to fill current vacancies in preparation for the future need. Currently, 30% of approved nursing positions are still vacant [28]. Although greater enrollment may be the solution, it may not be sufficient since some students admitted to nursing education drop out or leave the profession after graduation upon realizing that nursing is not meant for them [29].

There are limited studies available in Uganda to know the current generations' attitudes and perceptions towards the nursing profession. Hence, this study aimed to answer the following research questions: First, what are the attitudes of undergraduate nursing students towards the nursing profession and second, what the perceptions of undergraduate nursing students about the nursing profession?

## Methods

### Study design

An equal-status mixed methods design was used.

### Study setting and period

The study was carried out in two accredited public Universities in Uganda, Busitema University Faculty of Health Sciences and Makerere University College of Health Sciences, in the period between 20/11/2021 and 22/01/2022.

Busitema University Faculty of Health Sciences is located within Mbale Regional Referral Hospital along Pallisa Road, Mbale City, Eastern Uganda approximately 220 kilometers northeast of Kampala, the capital city of Uganda. The faculty was established in 2013. The Bachelor of Science in Nursing program was introduced in 2015 and has an average enrolment of 22 students per intake.

Makerere University College of Health Sciences is a semi-autonomous college of Makerere University located on Mulago Hill between the new and old Mulago Hospital in northeast Kampala, the capital city of Uganda. The Bachelor of Science in Nursing Program was introduced in 1993 in the then Faculty of Medicine and has an average enrolment of 25 students per intake.

### Study population

Bachelor of Science in Nursing students from first to fourth year at the two universities

### Sampling strategy

All nursing students in the two universities were invited to participate in the census survey in the quantitative arm of the study since they were only 180 in number.

Participants for the qualitative arm were obtained by purposive sampling. Two students were selected from each year of study considering the gender and entry scheme of the student.

## Data collection methods and tools

**Quantitative data collection.** A structured self-administered questionnaire was utilized consisting of three sections. The socio-demographic section included the independent variables; age, gender, religion, marital status, home area, the institution of study, year of study, entry scheme, father's level of education, and source of funding for the academic program. The second section was developed to assess the students' ranking of the nursing profession and their plans after graduation based on a literature review of published studies [13, 21, 23].

The third section about attitudes was adopted from the Attitude Scale for the Nursing profession (ASNP) developed by Coban and Kasikci in 2011, consisting of 40 items. The attitude subscales are; properties of the nursing profession (1–18), preference to the nursing profession (19–31), and general position of the nursing profession (32–40). It is a five-point Likert-type scale with a minimum score of 40 and maximum of 200. A total score of 120 and above indicates a positive attitude towards the profession [30]. For this study, a 4-point Likert scale was used where 1 signified strongly disagree and 4 signified strongly agree. Cronbach's Alpha internal consistency coefficient of the scale is 0.91.The Cronbach's Alpha coefficient for this study was 0.79.

Participant recruitment occurred during study sessions where the researcher distributed questionnaires after providing information about the study. Participants were allowed time to fill and return filled questionnaires to the researcher. In some classes, class leaders collected filled questionnaires and delivered them to the researcher. No one involved in student teaching or assessment was involved in recruitment or data collection. The study response rate was 91.6%.

A pilot study was carried out involving 15% (n = 24) of nursing students from Soroti University however the response rate was 33.3%.

**Qualitative data collection.** Qualitative data was collected using a structured interview guide in two focus group discussions conducted in English by trained and experienced researchers. Participants voluntarily provided verbal informed consent before the start of the discussion. Each discussion involved a representative sample of eight participants both male and female, direct entry and diploma entry students from each year of study. Audio recordings were put in place to gather all the data. The standard of focus group discussions was data saturation. To ensure validity, reliability and worthiness, the researcher provided clear information to the participants, established a relationship based on trust, accurately recorded and compared findings between the two groups. The participants chosen were able to provide relevant data to the study since they were knowledgeable and willing to respond.

## Data analysis

**Quantitative data analysis.** Quantitative data was cleaned and entered into an excel sheet which was exported to Stata version 15 statistical software for analysis. Univariate findings were described using frequency, proportions, and measures of central tendency.

Association between attitudes of participants was assessed using the Kruskal-Wallis one-way analysis of variance and t-test. A $p$-value of $<0.05$ was considered statistically significant.

**Qualitative data analysis.** Qualitative data was analyzed by thematic analysis as follows; Data collected was transcribed from audio recording to text and compiled into transcripts that were assigned to another researcher who was a co-coder. Before coding, the researcher read through the scripts to fully understand what was discussed.The transcripts were coded based

on phrases, sentences, and paragraphs [31]. The two researchers discussed and generated a final list of codes. Similar codes were identified and these were used to generate themes. Emerging themes were presented as a percentage of students in each theme. Verbatim quotes were also presented to back up the themes.

### Ethics statement

Ethical clearance was obtained from Mbale Regional Referral Hospital Review Ethical Committee, IRB approval number MRRH-2021-89. Administrative clearance was obtained from the Dean of Makerere University College of Health Sciences to carry out the study at Makerere University. The respondents provided written informed consent after providing information about the study. Participants were assured of confidentiality, their name and registration numbers were not included anywhere in the questionnaire.

## Results

### Quantitative results

**Socio-demographic characteristics of the nursing students.** Out of the 165 participants who responded to the study, 86 (52.4%) were female. One hundred and ten (66.7%) were in the age group 20–24 years with a mean age of 25.3±5.5 years (minimum 20, maximum 60). The majority 103 (80.6%) were single and 147 (89.1%) were Christians. Ninety-five participants (57.6%) were from urban areas and their fathers had attained tertiary education (55.2%). Eighty-four students (50.9%) were from Busitema University. Forty-seven participants (28.5%) were in the third year of study. One hundred twenty-four participants (75.2%) joined through direct entry scheme. Furthermore, fifty-six participants (33.9%) obtained financial support to pursue the program through government sponsorship (Table 1).

**Nursing students' ranking of the nursing profession and plans after graduation.** Seventy-four participants (44.9%) ranked the nursing profession second choice, followed by sixty-seven (40.6%) ranking it first choice. Seventy-four participants (44.9%) plan for higher education, followed by 21.8% for nursing administration (n = 36), 15.8% for teaching institution (n = 26), while 19 (11.5%) plan to change careers and only 10 (6.1%) plan to join bedside nursing after graduation.

**Attitudes of nursing students towards the nursing profession.** One hundred thirty-five students (81.8%) had positive attitudes towards the Nursing profession. The mean attitude score was 129.9±12.3 (minimum 87, maximum 153). Items in "Properties of the Nursing profession" attracted the highest mean score of 61.9(SD 5.9) and items in the General position of the Nursing profession had the lowest mean score of 27.5(SD 2.6) (Table 2).

When results of the t-test were examined, there were significant differences in attitudes according to the year of study and entry scheme. ($R2 = 0.12$, F = 2.21, p = 0.01). The year of study and entry scheme explained 12% of the total variance in attitudes of the students (Table 3).

The father's education level and the source of funding did not have influence on the nursing students' attitudes towards the nursing profession (Table 4).

### Qualitative results

Out of the sixteen participants for the focus group discussions, 8(50%) were female. Eight (50%) were from Busitema University and nine (56.25%) were aged between 20–24 years.

**Table 1. Socio-demographic characteristics of the nursing students.**

| Variable | Categories | Frequency (n = 165) | Percentage (%) |
|---|---|---|---|
| **Age** | 20–24 | 110 | 66.7 |
| | ≥25 | 55 | 33.3 |
| **Gender** | Female | 86 | 52.1 |
| | Male | 79 | 47.9 |
| **Marital status** | Married | 32 | 19.4 |
| | Single | 133 | 80.6 |
| **Home area** | Rural | 70 | 42.4 |
| | Urban | 95 | 57.6 |
| **Religion** | Christian | 147 | 89.1 |
| | Moslem | 18 | 10.9 |
| **Institution of study** | Busitema University | 84 | 50.9 |
| | Makerere University | 81 | 49.1 |
| **Year of study** | Year 1 | 42 | 25.5 |
| | Year 2 | 32 | 19.4 |
| | Year 3 | 47 | 28.5 |
| | Year 4 | 44 | 26.7 |
| **Entry scheme into university** | Direct entry scheme | 124 | 75.2 |
| | Diploma entry scheme | 41 | 24.9 |
| **Father's education level** | Illiterate | 10 | 6.1 |
| | Primary | 26 | 15.8 |
| | Secondary | 38 | 23.0 |
| | Tertiary | 91 | 55.2 |
| **Source of funding** | Government sponsored | 56 | 33.9 |
| | NGO | 32 | 19.4 |
| | Parents/ Guardian | 48 | 29.1 |
| | Self | 29 | 17.6 |

**Choice of the nursing profession.** Nursing students had numerous reasons for choosing the nursing profession. Three participants (18.75%) chose nursing because of interest and considered it as their first choice.

*"I had the interest in nursing so I applied for it without applying for other courses and indeed I was taken on merit because. It was my favorite." (APN2)*

Three participants (18.75%) chose nursing based on their academic performance. Some participants were academically weak and their admission to study nursing was a privilege, a divine intervention, which would not have been possible without God.

*"I used to pray to God to give me any medical course be it Medicine, Nursing, Anesthesia, because I was not very good at school so I knew my weakness. When I joined, many people were telling me to change the course because it is not good but, I know myself." (APN3)*

**Table 2. Mean attitude scores of nursing students obtained from the ASNP.**

| ASNP Subscale | Mean | SD | Minimum | Maximum |
|---|---|---|---|---|
| Properties of the Nursing profession | 61.9 | 5.9 | 35 | 71 |
| Preference for the Nursing profession | 40.4 | 6.5 | 20 | 52 |
| General position of the Nursing profession | 27.5 | 2.6 | 19 | 34 |

**Table 3.  Analysis of mean attitudes according to socio-demographic variables using t-test.**

| Variable | Coefficient | SE | t | P>\|t\| | [95% Conf. Interval] |
|---|---|---|---|---|---|
| Age | -0.115 | 0.0965 | -1.20 | 0.23 | -0.306 0.075 |
| Gender | -0.015 | 0.0622 | -0.25 | 0.80 | -0.138 0.107 |
| **Year of study** | **-0.160** | **0.0629** | **-2.55** | **0.01** | **-0.284 -0.036** |
| Religion | 0.118 | 0.0961 | 1.23 | 0.22 | -0.071 0.308 |
| Marital status | 0.047 | 0.1209 | 0.40 | 0.69 | -0.191 0.286 |
| Home area | -0.046 | 0.063 | -0.74 | 0.46 | -0.171 0.078 |
| **Entry scheme** | **0.274** | **0.1355** | **2.02** | **0.04** | **0.006 0.541** |
| Institution of study | -0.112 | 0.060 | -1.87 | 0.06 | -0.231 0.006 |

Observations = 165 R2 = 0.1256 F = 2.212 p = 0.0198

Seven participants (43.75%) had mixed reasons for choosing the nursing profession such as, availability of the course, admission on merit, low cut-off points, parents' choice, and desire to pursue a course from a specific institution. Of these, the desire to pursue a course at a particular institution dominated and was the main reason for their motivation to study nursing.

*"I did not decide to do nursing myself; I was given nursing at the University. Also, since the admission document showed that I was given nursing, it was the best option at the moment. I got converted and had to pursue it since I wanted a course from Makerere." (APN9)*

*"I wanted to deal with human beings but still I only wanted to study at Busitema University. Looking at the cut-off points, I knew nursing would be assured so I applied for only nursing at Busitema." (APN7)*

Three participants (18.75%) were straightforward about choosing the nursing profession because it was the last option after failing to get admitted for Medicine and Surgery. At the time of admission, these participants were not happy but later accepted and found comfort in the fact that nursing is also a profession in the health sector.

*"When I was studying, my interest was to do medicine and surgery but many times things do not turn out the way you expect so since I failed to get admitted for medicine and surgery, I resorted to joining. In a nutshell, nursing was the last resort to entering the medical field." (APN8)*

**Perceptions of nursing students about the nursing profession.**   Perceptions of the participants about the nursing profession are shown in Table 5.

**Table 4.  Analysis of mean attitudes according to socio-demographic variables using Kruskal- Wallis test.**

| Variable | Categories | Observations | Rank sum | kW | P-value |
|---|---|---|---|---|---|
| Father's education level | Illiterate | 10 | 732.50 | 1.033 | 0.793 |
| | Primary | 26 | 2218.00 | | |
| | Secondary | 38 | 2981 | | |
| | Tertiary | 91 | 7763.00 | | |
| Source of funding | Government | 56 | 4415.50 | 4.174 | 0.2432 |
| | NGO | 32 | 2888.50 | | |
| | Parents/Guardian | 48 | 3631.50 | | |
| | Self | 29 | 2759.50 | | |

**Table 5. Themes on perceptions of nursing students about the nursing profession.**

| Themes | Yes Frequency (%) | No Frequency (%) |
|---|---|---|
| Nursing is a good profession that provides opportunity to interact with patients | 9 (56.25%) | 7 (43.75%) |
| The image of the nursing profession | 12(75.0%) | 4 (25.0%) |
| Nursing is a hectic and traumatizing profession | 10 (62.5%) | 6 (37.5%) |
| Nurses are not allowed to practice what they learn at school | 4 (25%) | 12 (75%) |
| Bias towards employing lower cadres in nursing compared to graduate nurses | 12 (75%) | 4 (25%) |
| Leadership in nursing | 7 (43.75%) | 9 (56.25%) |
| Poor remuneration of nurses | 13 (81.25%) | 3 (18.75%) |

*Nursing is a good profession that provides an opportunity to interact with patients*. Overall, the respondents perceived nursing as a good and enjoyable profession, which imparts knowledge needed to care for patients and manage illnesses. Nursing facilitates interaction with patients as nurses are the first caregivers and spend more time with patients.

> *"It has been my favorite, am proud of it and want to explore more about how lives are saved, and how to care for the sick. To me it is good because we are working to take care of the patient, irrespective of what one is doing, we are all geared towards a common goal." (APN2)*

A great number of negative perceptions were identified as presented below;

*The image of the nursing profession*. Nursing students had a very poor image of the nursing profession. The participants saw no specific role they play in practice and would not advance with a career in nursing because of fear to progress and yet remain nothing.

> *"There is a problem with the people in nursing; lack of creativity and leaning more on history. We keep on looking at people with ancient ideas and yet they decide what we have to practice so we are always rotating around the same axis. There are things, which should change such as the way senior nurses expect to be treated."* (APN9)

The participants perceived nursing as a profession that is not popular among the public.

> *"People are not sensitized enough about this profession, when you say you're a nurse at Bachelor's level, someone wonders what you do. People are discouraged from joining the profession. With time students will even stop enrolling into the profession because of fear of what they are going to go through." (APN10)*

They narrated that the various stakeholders marginalize the nursing profession.

> *"It is very unfortunate that people do not appreciate or recognize you. Respective authorities marginalize the nursing profession. This is expected because this seems to be a female-dominated profession and anything that is comprised of women, people under look it." (APN6)*

The participants stated that other health professional students, educators, and senior nurses minimized them. The senior nurses tend to respect other health professional students and treasure them while looking down on nursing students.

*"As a student, am under looked by physicians on the ward and wherever we are. The medical students look at us like a class lower than them. Even the nurses, our colleagues already in the field consider students of other courses to be more important than us." (APN5)*

*Nursing is a hectic and traumatizing profession.* Nursing was described as a very hectic profession where nurses spend a lot of time with patients and lack time for themselves and their families. The profession was labeled one that exposes nurses to vast challenges ranging from personal safety threats to torture. Bedside nurses and nursing students are more vulnerable to torture.

*"Bedside nursing is one of the most hectic activities that a nurse has to undergo because it is a close relationship between you and the patient and you always need to do the needful. This ranges from turning the patient, wound dressing. You always have to be on standby." (APN2)*

*"Nursing students, if you're not strong you may even leave the profession. There is a feeling of a minority like when a person tells you, "Oh you are doing nursing, sorry for you" so it creates an inferiority complex, someone feels they will not succeed in this world." (APN4)*

*Nurses are not allowed to practice what they learn at school.* The nursing profession was displayed to have restrictions about what a nurse can and cannot do, although nursing students are equipped with information, skills, and confidence to practice during the training. Additionally, most nurses upon qualification choose to work in the clinical setting and neglect the other fields of nursing such as research and leadership in nursing.

*"At the end of it all, what you are taught is not what you are going to practice. It may end in school, you are not going to clerk a patient, you are just going to be looking through the notes a doctor has made." (APN8)*

Bias towards employing lower cadres in nursing compared to the graduate nurses.

*"You're studying but people are telling you, where will you get a job, private sector employs certificate nurses, only the government employs bachelor nurses, who will connect you to get a government job. You need money to get a government job, where will you get the money." (APN4)*

*Leadership in nursing.* The participants described the leadership in nursing as a broken leadership that does not fight for the rights and safety of nurses. The nursing leaders as stated do not mentor the young generation and instead make decisions that lead to the suffering of members of the profession.

*"If there is even a chance to fight for their rights, you know every human being must come out to fight for their rights but sometimes they tend to say such statements, 'For me, I was called to love and serve' which is okay but if something needs action, you should come and act." (APN9)*

*Poor remuneration of the nurses.* Additionally, the participants expressed that the pay nurses receive is too little and not realistic yet they do a lot of work.

*"Jobs are there but is the juice worth the squeeze? Is the pay worth the work you are going to do, if someone has seen that going the other side is a better place for them, why settle for less somewhere where you are not even appreciated?"(APN9)*

*"Many of us after finishing opt for other fields outside nursing just because we know we shall be underpaid yet we have invested a lot to study. Now, we are nursing students but at the end of it all, we are going to join the nurses out there. If those people are crying, do you think we shall yearn to join them and also cry, no." (APN10)*

## Discussion

### Attitudes of nursing students towards the nursing profession

Majority (81.8%) of the students had positive attitudes towards the Nursing profession. This finding was higher than that reported by another study carried out at Hue University of Medicine and Pharmacy, Vietnam in which only 62.07% of the students had favorable attitudes towards the nursing profession [11]. This may be due to the influence of family and friends on the students who decided to study nursing yet they had no aspirations and interest to practice in hospital. In contrast, the students in this study had interest in nursing among other reasons such as academic performance. Therefore, influence from friends and family was not the main reason they decided to join the nursing profession.

Items in "Properties of the Nursing profession" attracted the highest mean score and items in the General position of the Nursing profession had the lowest mean score. This is consistent with a study carried out in Western Turkey [14] where characteristics of the nursing profession attained the highest mean score and the general position of the nursing profession the lowest mean score.

Properties of the nursing profession create an impression of nursing as a noble profession centered around education, advocacy, and care for patients [32, 33]. On the other hand, the low mean score of attitudes towards the general position of the nursing profession can be explained by the public view of nursing as a troubled profession that is not independent [34, 35] which creates a feeling of inferiority among the nursing students.

The current study revealed a significant difference in attitudes according to the year of study. Attitudes of nursing students towards the nursing profession vary as they progress from preclinical to clinical years of study, becoming more favorable [14] or less favorable [36] with increasing years in school. Students in clinical years of study undergo a hectic clinical practicum and stressful experiences such as being under looked by doctors and senior nurses on the wards. Nursing students sometimes have high expectations when joining the university, which are shattered as they encounter real experiences in the profession, which require difficult interventions.

Nursing students who join the university through the diploma entry scheme have prior exposure to nursing practice and knowledge about the profession and therefore have more interest and satisfaction with the profession compared to direct entry students whose view of the nursing profession depends on experiences and advice from others hence a significant difference in attitudes between the two groups.

### Perceptions of nursing students about the nursing profession

Students join nursing education with different ideas about nursing due to limited public knowledge about the nursing profession [37]. Participants in this study joined with perceptions of nursing as a weak course. Occasionally, such preconceived perceptions about nursing are not realistic and therefore altered by the reality of practice during or after the course of study.

In clinical practice, the roles of a nurse are limited to administering prescribed drugs, damp dusting and patient monitoring. Nurses are not allowed to prescribe drugs and have to wait for

a doctor to make the prescription. This causes frustration especially among graduate nurses, who feel they should play a bigger role than they do according to their scope of practice.

Nursing students ranked the nursing profession second choice. These results are in agreement with a similar study carried out in Egypt and Jordan where the students ranked the nursing profession second after the medical profession [38]. This highlights the reality that most of the students enroll for nursing after failing to meet the minimum requirements for Bachelor of Medicine and Surgery, which is usually the first choice. Some students join nursing because of lack of other opportunities while others are forced to join the profession [39] because of anticipated availability of jobs. Interest and motivation can be promoted by providing access to the right information and knowledge about the profession [40].

Only 6.1% of nursing students planned to join bedside nursing after graduation. This can be explained by the students' description of bedside nursing as the most hectic experience, associated with poor working conditions, unfair treatment, lack of time for personal responsibilities and family, and low pay. Therefore, nursing students leave the profession in search of opportunities with better pay, where they get value for the time they spend working in comfortable conditions. Similar studies have reported that nursing students were reluctant to join bedside nursing [41] especially because of negative experiences during clinical areas leading many of them to exit the profession.

The participants in this study expressed that nurses' pay is too little and not proportionate with the work they do. These findings are in agreement with a study [42] where 72% of nurses wanted to leave the profession because of low salaries. Proper remuneration improves nurses' performance and motivation. More employees are attracted to join and get retained in the profession [43].

The ratio of graduate nurses to lower cadre nurses in practice is low, partly because graduates are discouraged because of the low pay but most commonly because the employers prefer to employ nurses with certificate and diploma levels of qualification as opposed to graduates. Non-Government Organizations, teaching institutions and research centers, employ graduate nurses yet patients in the general healthcare system would greatly benefit from their critical thinking and specialized skills.

## Conclusion

The results of this study showed that the attitudes of undergraduate nursing students towards the nursing profession were positive but their perceptions about the nursing profession were negative. This provides an opportunity for nursing training institutions to develop interventions purposed to change the negative perceptions of nursing students about their profession. This will enable students to choose the nursing profession willingly and experience a higher level of satisfaction with their practice as a nurse after graduation.

## Limitations to the study

The findings from this study cannot be generalized because of the small sample size.

Although the new tool used in this study demonstrated reasonable internal consistency, further construct validation with a larger cohort is required.

## Implications of this study

Nursing training institutions should ensure that nursing students acquire and maintain an accurate perception of the nursing profession.

## Recommendations

An interventional study is recommended to facilitate a more positive change among nursing students. The study also recommends a four-year longitudinal study to determine how the attitudes and perceptions change over time.

## Acknowledgments

We acknowledge support from the Department of Nursing of Busitema University, Department of Nursing of Makerere University, Uganda Nurses and Midwives Council and our dear participants who gave us their honest views about the nursing profession.

## Author Contributions

**Conceptualization:** Irene Mildred Neumbe, Lydia Ssenyonga, David Jonah Soita, Rebecca Nekaka.

**Data curation:** Lydia Ssenyonga, Jacob Stanley Iramiot, Rebecca Nekaka.

**Formal analysis:** Irene Mildred Neumbe, David Jonah Soita, Rebecca Nekaka.

**Funding acquisition:** Rebecca Nekaka.

**Investigation:** Irene Mildred Neumbe, David Jonah Soita, Rebecca Nekaka.

**Methodology:** Irene Mildred Neumbe, Lydia Ssenyonga, David Jonah Soita, Rebecca Nekaka.

**Project administration:** Irene Mildred Neumbe, Lydia Ssenyonga.

**Supervision:** Lydia Ssenyonga, David Jonah Soita, Rebecca Nekaka.

**Validation:** Rebecca Nekaka.

**Writing – original draft:** Irene Mildred Neumbe, Lydia Ssenyonga, David Jonah Soita.

**Writing – review & editing:** Lydia Ssenyonga, David Jonah Soita, Jacob Stanley Iramiot, Rebecca Nekaka.

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
