## [Decision Letter · Decision Letter 0]

6 Apr 2023

PONE-D-23-00352Choice, Attitudes, and Perceptions of Undergraduate Nursing Students towards the Nursing ProfessionPLOS ONE

Dear Dr. Iramiot,

Thank you for submitting your manuscript to PLOS ONE. After careful consideration, we feel that it has merit but does not fully meet PLOS ONE’s publication criteria as it currently stands. Therefore, we invite you to submit a revised version of the manuscript that addresses the points raised during the review process.

We look forward to receiving your revised manuscript.

Kind regards,

Joyce Jebet Cheptum

Academic Editor

PLOS ONE

Journal Requirements:

Additional Editor Comments:

This is well written paper that highlights choices, attitudes and perceptions of nursing students towards nursing profession.

The methodology, especially for the qualitative section needs to be improved to include reliability , validity and trustworthiness. Ae you sure a pilot study was done? If so, this is a mini study and their findings should also be published.

What was the response rate? How many students were enrolled per class? In the methodology, preferably mention proportionate sampling , if it was done

Reviewers' comments:

Reviewer's Responses to Questions

**Comments to the Author**

1. Is the manuscript technically sound, and do the data support the conclusions?

Reviewer #1: Yes

Reviewer #2: Partly

2. Has the statistical analysis been performed appropriately and rigorously? 

Reviewer #1: Yes

Reviewer #2: No

3. Have the authors made all data underlying the findings in their manuscript fully available?

Reviewer #1: Yes

Reviewer #2: Yes

4. Is the manuscript presented in an intelligible fashion and written in standard English?

Reviewer #1: Yes

Reviewer #2: Yes

5. Review Comments to the Author

Reviewer #1: The manuscript describes the context and approach of the study and is conducted in a rigorous manner. The data provided supports the conclusions. The statistical analysis if rigorously conducted and presentation is appropriate for the type of data being presented. The authors have provided a link for the data access for scrutiny. It is written in standard English with very few grammatical errors. The language is clear. The manuscript will contribute to this area of scientific research if published in the website. The authors have consulted a wide range of current works that has been publish to support the study and findings.

Reviewer #2: Dear authors,

Thank you for giving me such great opportunity to review this manuscript.

In general, the topic is interesting one and there is a much effort undertaken. I have some inquires and some comments that will strengthen your manuscript.

I need to know why researchers conduct this study in Uganda. Numerous studies conducted and published concerning attitudes and perceptions of nursing students toward nursing profession. Also, factors shaping these attitudes and perceptions were also investigated too much. Results of the current study are the case of the vast majority of studies concerning attitudes and perceptions of students toward nursing profession. Is the context of nursing profession in Uganda is different?- what are the factors that make researchers to address this issue now in Uganda despite availability of solid background regarding attitudes and perceptions of different stakeholders of healthcare toward nursing profession.

This manuscript is too long and need major revision.

I know the effect of mixed research design in writing manuscript but this manuscript could be focused and summarized.

Please follow the following comments;

Abstract:

- Sampling technique and timing of data collection need to be given.

Introduction:

- Context of nursing profession in Uganda should be reflected enough ( regulation of profession, admission criteria, vital statistics, role of nursing in healthcare delivery system in Uganda)

- What about the attitudes and perceptions of students toward nursing profession from other regions?

- Conceptual framework not clear( on which base researchers develop this framework)

- Dependent and independent variables need to be addressed in introduction

- I find results about determinants of attitudes of students toward nursing which is not reflected clearly in the aim of study

- Why researchers ignore to add research question in this study?

Method:

- Why researchers choose the purposive sampling to select participants of focus group?

- Recruitment measures for participants should be mentioned - Sample representativeness, response rate is needed.

- Timing of conducting the study should be stated

- What are the measures used to ensure validity and trustworthy of data collected from subjects?

- What is the standard of focus group discussions; is it just point of view telling or data saturation?

- What about the structure of interview questions used in focus group discussion?

Results:

- Qualitative results need to be summarized ( many verbatim quotes not needed )

- I feel if quantitative analysis used to reflect percent of students in each theme of attitudes yielded from focus group, it will give powerful impression since researchers depend only on 16 nurses to give us results about perceptions and attitudes which is questionable

- Comparison of attitudes obtained from ASNP using mean scores did not give value to the readers. It is better to use mean score percent.

- What is the meaning of "relationship between students gender, marital status, religion, entry scheme and their attitudes was negligible but significant" ?

Discussion:

- It cannot find an interesting effective debate in discussion (researchers must keep their own voice too much using effective debate guidelines).

- Some results need to be discussed as why 6.1% only of students plan to join bedside nursing after graduation

Implications:

- What are the implications from this study for both clinicians and healthcare leaders?

6. PLOS authors have the option to publish the peer review history of their article (what does this mean?). If published, this will include your full peer review and any attached files.

Reviewer #1: **Yes: **Jane W Kabo

Reviewer #2: No

---

## [Author Response · Author response to Decision Letter 0]

29 Apr 2023

PONE-D-23-00352 Choice, Attitudes, and Perceptions of Undergraduate Nursing Students towards the Nursing Profession

Response to Reviewers

1. When submitting your revision, we need you to address these additional requirements. Please ensure that your manuscript meets PLOS ONE's style requirements, including file naming. 

Response: These additional requirements have been addressed

2. Please provide additional details regarding participant consent. In the ethics statement in the Methods and online submission information, please ensure that you have specified what type you obtained (for instance, written or verbal, and if verbal, how it was documented and witnessed). 

Response: Written consent was obtained. The Informed consent form had provision for the participants to acknowledge consent by appending a signature and the date.

3. Your ethics statement should only appear in the Methods section of your manuscript

Response: The ethics statement has been moved to the Methods selection on page 6.

Response: The reference list has been reviewed

Additional Editor Comments:

This is well written paper that highlights choices, attitudes and perceptions of nursing students towards nursing profession. The methodology, especially for the qualitative section needs to be improved to include reliability, validity and trustworthiness. 

Response: Reliability, validity and worthiness have been included in the qualitative methodology on page 6.

Are you sure a pilot study was done? If so, this is a mini study and their findings should also be published. What was the response rate? How many students were enrolled per class? In the methodology, preferably mention proportionate sampling , if it was done

Response: A pilot study was done. 12 students were enrolled, six per class however only four responded giving a response rate of 33.3%. I do not think their findings should be published. 

 Reviewer's Responses to Questions

Comments to the Author

Reviewer #1: The manuscript describes the context and approach of the study and is conducted in a rigorous manner. The data provided supports the conclusions. The statistical analysis if rigorously conducted and presentation is appropriate for the type of data being presented. The authors have provided a link for the data access for scrutiny. It is written in standard English with very few grammatical errors. The language is clear. The manuscript will contribute to this area of scientific research if published in the website. The authors have consulted a wide range of current works that has been publish to support the study and findings.

Reviewer #2: Dear authors, Thank you for giving me such great opportunity to review this manuscript. In general, the topic is interesting one and there is a much effort undertaken. I have some inquires and some comments that will strengthen your manuscript. I need to know why researchers conduct this study in Uganda. Numerous studies conducted and published concerning attitudes and perceptions of nursing students toward nursing profession. Also, factors shaping these attitudes and perceptions were also investigated too much. Results of the current study are the case of the vast majority of studies concerning attitudes and perceptions of students toward nursing profession. Is the context of nursing profession in Uganda is different?- what are the factors that make researchers to address this issue now in Uganda despite availability of solid background regarding attitudes and perceptions of different stakeholders of healthcare toward nursing profession.

Response: This study was conducted because of the low average enrolment of nursing students in Ugandan universities compared to other medical courses and incidences of nursing students changing study programs even after reaching third year of study.

The context of the nursing profession in Uganda is different, nursing students are equally important stakeholders and their views matter because they are directly associated with the profession compared to different stakeholders.

This manuscript is too long and need major revision. I know the effect of mixed research design in writing manuscript but this manuscript could be focused and summarized.

Response: The manuscript has been revised and shortened 

Please follow the following comments;

Abstract:

- Sampling technique and timing of data collection need to be given.

Response: The sampling technique and timing have been included

Introduction:

- Context of nursing profession in Uganda should be reflected enough ( regulation of profession,

admission criteria, vital statistics, role of nursing in healthcare delivery system in Uganda)

Response: The context of the nursing profession in Uganda has been incorporated in the background

- What about the attitudes and perceptions of students toward nursing profession from other regions?

Response: The attitudes and perceptions of students from other regions has also been included

- Conceptual framework not clear( on which base researchers develop this framework)

Response: The conceptual framework has been omitted to shorten the manuscript. However it consisted of independent variables; age, gender, marital status, home area, year of study, institution of study, entry scheme, father’s education level and source of funding. The dependent variables were choice, attitudes and perceptions of undergraduate nursing students towards the nursing profession.

- Dependent and independent variables need to be addressed in introduction

Response: The independent variables have been addressed in the methods section, quantitative data collection methods and tools.

- I find results about determinants of attitudes of students toward nursing which is not reflected clearly in the aim of study

Response: The results have been omitted

- Why researchers ignore to add research question in this study?

Response: Research questions have been added

Method:

- Why researchers choose the purposive sampling to select participants of focus group?

Response: The participants chosen purposively were willing and able to provide relevant data to the study since they were knowledgeable about the topic

- Recruitment measures for participants should be mentioned - Sample representativeness, response rate is needed.

Response: Recruitment measures have been mentioned

- Timing of conducting the study should be stated

Response: Timing of conducting the study has been stated

- What are the measures used to ensure validity and trustworthy of data collected from subjects?

Response: To ensure validity, reliability and worthiness, the researcher provided clear information to the participants, established a relationship based on trust, accurately recorded and compared findings between the two groups

- What is the standard of focus group discussions; is it just point of view telling or data saturation?

Response: The standard was data saturation

- What about the structure of interview questions used in focus group discussion?

Response: A structured interview guide was followed

Results:

- Qualitative results need to be summarized ( many verbatim quotes not needed )

Response: Qualitative results have been summarized

- I feel if quantitative analysis used to reflect percent of students in each theme of attitudes yielded from focus group, it will give powerful impression since researchers depend only on 16 nurses to give us results about perceptions and attitudes which is questionable

Response: Quantitative analysis has been used to reflect percent of students in each theme

- Comparison of attitudes obtained from ASNP using mean scores did not give value to the readers. It is better to use mean score percent.

Response: This comment is not well understood

- What is the meaning of "relationship between students gender, marital status, religion, entry scheme and their attitudes was negligible but significant" ?

Response: This statement was irrelevant and has been omitted.

Discussion:

- It cannot find an interesting effective debate in discussion (researchers must keep their own voice too much using effective debate guidelines).

Response: Noted.

- Some results need to be discussed as why 6.1% only of students plan to join bedside nursing after graduation

Response: These results have been discussed

Implications:

- What are the implications from this study for both clinicians and healthcare leaders?

Response: Clinical instructors and lecturers should create awareness programs to equip students with knowledge about the nursing and its importance, focusing on improving the image of the profession.

Health care leaders should advocate for better remuneration and employment of nurses. They should also emphasize and embrace specialty in nursing to increase efficiency and improve quality of service delivery.

---

## [Decision Letter · Decision Letter 1]

6 Jun 2023

PONE-D-23-00352R1Choice, Attitudes, and Perceptions of Undergraduate Nursing Students towards the Nursing ProfessionPLOS ONE

Dear Dr. Iramiot,

Thank you for submitting your manuscript to PLOS ONE. After careful consideration, we feel that it has merit but does not fully meet PLOS ONE’s publication criteria as it currently stands. Therefore, we invite you to submit a revised version of the manuscript that addresses the points raised during the review process.

We look forward to receiving your revised manuscript.

Kind regards,

Joyce Jebet Cheptum

Academic Editor

PLOS ONE

Journal Requirements:

Additional Editor Comments (if provided):

Please address comments from the third reviewer after the first revision of the initial document

Reviewers' comments:

Reviewer's Responses to Questions

**Comments to the Author**

1. If the authors have adequately addressed your comments raised in a previous round of review and you feel that this manuscript is now acceptable for publication, you may indicate that here to bypass the “Comments to the Author” section, enter your conflict of interest statement in the “Confidential to Editor” section, and submit your "Accept" recommendation.

Reviewer #2: All comments have been addressed

Reviewer #3: (No Response)

2. Is the manuscript technically sound, and do the data support the conclusions?

Reviewer #2: Yes

Reviewer #3: Partly

3. Has the statistical analysis been performed appropriately and rigorously? 

Reviewer #2: Yes

Reviewer #3: No

4. Have the authors made all data underlying the findings in their manuscript fully available?

Reviewer #2: No

Reviewer #3: No

5. Is the manuscript presented in an intelligible fashion and written in standard English?

Reviewer #2: Yes

Reviewer #3: No

6. Review Comments to the Author

Reviewer #2: Dear Authors Dear authors

I am more than pleased to review this manuscript

Thanks a lot for considering first review comments

This manuscript is now well coordinated, focused, and could be published in PLOS ONE

Absolutely, politicians, healthcare leaders and nursing leaders in Uganda will find a valid evidence to make a sound reform in nursing profession.

Again, thanks a lot for this fruitful paper that try to upgrade nursing profession in Africa countries.

Reviewer #3: Choice, Attitudes, and Perceptions of Undergraduate Nursing Students towards the Nursing Profession

Review

Comments to the Author

I noticed major issues that needs to be concerned

Title

The Title should be modified as remove as choice, “Attitudes and Perceptions of Undergraduate Nursing Students towards the Nursing Profession “ because they are already engagement of the profession.

Abstract

Background: please try to describe precise information about perception and attitude and finaly show the gap here

Methods: you describe only Attitude but what about perception related questionnaires, and please incorporate data analysis information in methodology part

Result: first write the prevalence then significant variable. what is your justification:- poor academic performance in high school, desire to pursue a medical course, failure to get admitted for Medicine and Surgery, low cut-off points for the course and interest in nursing.

Attitude result shows 81.8% what about perception and choice?

Conclusion: totally your conclusion wrong <pre-nurse about="" accurate="" acquisition="" an="" and="" counseling="" courses="" enhance="" introductory="" maintenance="" nursing="" of="" perception="" profession="" the="" to=""> they discuss not being included in your finding. Add please concise and short recommendation in the conclusion.

Background

Please describe in detail factors magnitude of attitude, perception in background.

Methods

Please replace study setting with study setting and period

Despite you used a self-administered survey that took two months to complete, your sample of 165 people is genuine

Study participants

Please write in the form of source population and study population than study participants.

Sampling strategy

Please write sampling strategy in clear and concise way. In general it is not clear the strategy

I'm not understanding how to take 160 samples please write with evidence based .

Your sample sizes are not adequate and dose not representative this small size sample. So it is difficult to accept your finding for scientific evidence purpose.

It is even possible to include other national-level universities.

Sampling procedure

Please write clearly show your sampling procedure in clear and specific schematic way

Quantitative data collection

For attitude type use a five-point Likert-type but what about perception.

Please write attitude and perception tools in a separated and clearly way

The response rate was 91.6%; what had happened prior to these results?

Your Pilot study samples size not be correct. Because 15% of your sample is12 replace by 24

Quantitative data analysis

Completely unclear data analysis procedure; please rewrite again; it would be wiser to start with a clean check and enter and so…..

Result

First the prevalence then the response rate in the result?

It is better to write majority 89.1% Christian and Busitema University students around half like (50%) and so….

Marital status and religion should have more than two add others (specify)

Father education level tertiary it need operationalized

Overall, the respondents perceived nursing as a good and enjoyable course (nursing is profession or course) not clear to me

Discussion

The first paragraph no need of write the objective here it is better to write general information about attitude and perception.

In the second paragraph these results are in agreement with a similar study in choice what is your justification in general your result look like possibility not real finding.

Attitudes of nursing students towards the nursing profession

The mean attitude score was 129.9±12.3 (minimum 87, maximum 153).but in data collection show that It is a five-point Likert-type scale with a minimum score of 40 and maximum of 200)

Before compare your result first writes the prevalence and range of attitude

Your study prevalence 81.8% had positive attitudes towards the nursing profession then how 62.07% similar Hue University of Medicine and Pharmacy need clarification

In general the whole discussion need detail description compare and contract

Conclusion

Your conclusion part not be focus your result. Please avoid terms like receive pre-nurse counseling and introductory courses about the profession, to enhance acquisition and maintenance of an accurate perception related variable, and conclude in line with your finding.

Limitations to the study

The findings cannot be generalized private but also the governmental nursing institutions in Uganda, because your sample size is very small.

Implications of this study

In general the implication of your study not based on your results

Reference

Most of the reference out dated eg Ref 14,15,23 ….

In general, I am interested in paper and recommend publication with major modifications.</pre-nurse>

7. PLOS authors have the option to publish the peer review history of their article (what does this mean?). If published, this will include your full peer review and any attached files.

Reviewer #2: No

Reviewer #3: No

---

## [Author Response · Author response to Decision Letter 1]

6 Jul 2023

Choice, Attitudes, and Perceptions of Undergraduate Nursing Students towards the Nursing Profession Review 

Comments Response to comments Reference page

Title 

The Title should be modified as remove as choice, “Attitudes and Perceptions of Undergraduate Nursing Students towards the Nursing Profession “ because they are already engagement of the profession.

 The title has been modified to remove choice Page 1

Abstract 

Background: please try to describe precise information about perception and attitude and finally show the gap here 

 The perception and attitudes have been described precisely Page 1

Methods: you describe only Attitude but what about perception related questionnaires, and please incorporate data analysis information in methodology part

 The ASNP is used to measure attitudes towards the nursing profession, it does not measure perceptions and choice. 

Result: first write the prevalence then significant variable. what is your justification:- poor academic performance in high school, desire to pursue a medical course, failure to get admitted for Medicine and Surgery, low cut-off points for the course and interest in nursing. 

Attitude result shows 81.8% what about perception and choice? 

 The results for perceptions are qualitative Page 1

Conclusion: totally your conclusion wrong <pre-nurse counseling and introductory courses about the profession to enhance the acquisition and maintenance of an accurate perception of the nursing profession> they discuss not being included in your finding. Add please concise and short recommendation in the conclusion. A concise recommendation has been added to the conclusion Page 1

Background 

Please describe in detail factors magnitude of attitude, perception in background. This has been considered Page 2

Methods 

Please replace study setting with study setting and period 

Despite you used a self-administered survey that took two months to complete, your sample of 165 people is genuine Study setting has been replaced with study setting and period

The sample size was the total number of nursing students in the two universities. Page 4

Study participants 

Please write in the form of source population and study population than study participants. Study participants has been replaced with study population Page 4

Sampling strategy

Please write sampling strategy in clear and concise way. In general it is not clear the strategy 

I'm not understanding how to take 160 samples please write with evidence based.

Your sample sizes are not adequate and dose not representative this small size sample. So it is difficult to accept your finding for scientific evidence purpose. 

It is even possible to include other national-level universities The sample size was the total number of nursing students in the two universities.

All the students were invited to respond to the questionnaire because of the small number.

 Page 4

Sampling procedure 

Please write clearly show your sampling procedure in clear and specific schematic way 

 For the quantitative arm, all students were invited to fill the questionnaire, they were not sampled

Purposive sampling was used for the Focus Group Discussions only Page 4

Quantitative data collection

For attitude type use a five-point Likert-type but what about perception. 

Please write attitude and perception tools in a separated and clearly way 

The response rate was 91.6%; what had happened prior to these results?

Your Pilot study samples size not be correct. Because 15% of your sample is12 replace by 24 Perceptions were collected through the Focus Group Discussions. They were qualitative findings of the study

Pilot study sample size has been corrected Page 5

Quantitative data analysis

Completely unclear data analysis procedure; please rewrite again; it would be wiser to start with a clean check and enter and so….. The data analysis procedure has been clearly described Page 6

Result 

First the prevalence then the response rate in the result?

It is better to write majority 89.1% Christian and Busitema University students around half like (50%) and so….

Marital status and religion should have more than two add others (specify)

Father education level tertiary it need operationalized 

Overall, the respondents perceived nursing as a good and enjoyable course (nursing is profession or course) not clear to me The first section is simply describing the socio-demographics of the participants

There was no other religion noted in the responses, except denominations of Christianity such as Anglicans and Catholics

Nursing is a profession not a course 

Discussion 

The first paragraph no need of write the objective here it is better to write general information about attitude and perception. 

In the second paragraph these results are in agreement with a similar study in choice what is your justification in general your result look like possibility not real finding. 

Attitudes of nursing students towards the nursing profession

The mean attitude score was 129.9±12.3 (minimum 87, maximum 153).but in data collection show that It is a five-point Likert-type scale with a minimum score of 40 and maximum of 200)

Before compare your result first writes the prevalence and range of attitude 

Your study prevalence 81.8% had positive attitudes towards the nursing profession then how 62.07% similar Hue University of Medicine and Pharmacy need clarification 

In general the whole discussion need detail description compare and contract The minimum score one can obtain using the ASNP is 40 and the maximum is 200 but for this study, the minimum score obtained was 87 and the maximum was 153. 

Conclusion 

Your conclusion part not be focus your result. Please avoid terms like receive pre-nurse counseling and introductory courses about the profession, to enhance acquisition and maintenance of an accurate perception related variable, and conclude in line with your finding. The conclusion has been improved Page 15

Limitations to the study

The findings cannot be generalized private but also the governmental nursing institutions in Uganda, because your sample size is very small. This limitation has been included Page 15

Implications of this study

In general the implication of your study not based on your results 

Reference 

Most of the reference out dated eg Ref 14,15,23 ….

In general, I am interested in paper and recommend publication with major modifications. Reference 14 is The Uganda Nurses and Midwives Act,1996 which provides for the training, registration, enrollment and discipline of nurses and midwives in Uganda. It was enacted in 1996 and has not been reviewed since then.

Reference 23 is the Attitude Scale for the Nursing Profession which was developed in 2011.

---

## [Editor Report · Decision Letter 2]

10 Jul 2023

Attitudes, and Perceptions of Undergraduate Nursing Students towards the Nursing Profession

PONE-D-23-00352R2

Dear Dr. Iramiot,

We’re pleased to inform you that your manuscript has been judged scientifically suitable for publication and will be formally accepted for publication once it meets all outstanding technical requirements.

Kind regards,

Joyce Jebet Cheptum

Academic Editor

PLOS ONE
---

## [Editor Report · Acceptance letter]

17 Jul 2023

PONE-D-23-00352R2 

Attitudes and Perceptions of Undergraduate Nursing Students towards the Nursing Profession 

Dear Dr. Iramiot:

I'm pleased to inform you that your manuscript has been deemed suitable for publication in PLOS ONE. Congratulations! Your manuscript is now with our production department. 

Kind regards, 

on behalf of

Dr. Joyce Jebet Cheptum 

Academic Editor

PLOS ONE